# Second-line treatment in advanced gastric cancer: Data from the Spanish AGAMENON registry

**Almudena Cotes Sanchís[1], Javier Gallego[2]\*, Raquel Hernandez[3], Virginia Arrazubi[4], Ana Custodio[5], Juana María Cano[6], Gema Aguado[7], Ismael Macias[8], Carlos Lopez[9], Flora López[10], Laura Visa[11], Marcelo Garrido[12], Nieves Martínez Lago[13], Ana Fernández Montes[14], María Luisa Limón[15], Aitor Azkárate[16], Paola Pimentel[17], Pablo Reguera[18], Avinash Ramchandani[19], Juan Diego Cacho[20], Alfonso Martín Carnicero[21], Mónica Granja[22], Marta Martín Richard[23], Carolina Hernández Pérez[24], Alicia Hurtado[25], Olbia Serra[26], Elvira Buxo[27], Rosario Vidal Tocino[28], Paula Jimenez-Fonseca[29], Alberto Carmona-Bayonas[30]**

1 Medical Oncology Department, Hospital General Universitario de Elda, Alicante, Spain, 2 Medical Oncology Department, Hospital General Universitario de Elche, Elche, Spain, 3 Medical Oncology Department, Hospital Universitario de Canarias, Tenerife, 4 Medical Oncology Department, Complejo Hospitalario de Navarra, Pamplona, Spain, 5 Medical Oncology Department, Hospital Universitario La Paz, CIBERONC CB16/12/00398, Madrid, Spain, 6 Medical Oncology Department, Hospital General Universitario de Ciudad Real, Ciudad Real, Spain, 7 Medical Oncology Department, Hospital Universitario Gregorio Marañón, Madrid, Spain, 8 Medical Oncology Department, Hospital Universitario Parc Tauli, Sabadell, Spain, 9 Medical Oncology Department, Hospital Universitario Marqués de Valdecilla, IDIVAL, Santander, Spain, 10 Medical Oncology Department, Hospital Universitario Doce de Octubre, Madrid, Spain, 11 Medical Oncology Department, Hospital Universitario El Mar, Barcelona, Spain, 12 Medical Oncology Department, Pontificia Universidad Católica de Chile, Santiago de Chile, Chile, 13 Medical Oncology Department, Complejo Hospitalario Universitario de A Coruña, Coruña, Spain, 14 Medical Oncology Department, Complejo Hospitalario de Orense, Orense, Spain, 15 Medical Oncology Department, Hospital Universitario Virgen del Rocío, Sevilla, Spain, 16 Medical Oncology Department, Hospital Universitario Son Espases, Mallorca, Spain, 17 Medical Oncology Department, Hospital Santa Lucía, Cartagena, Spain, 18 Medical Oncology Department, Hospital Universitario Ramón y Cajal, Madrid, Spain, 19 Medical Oncology Department, Hospital Universitario Insular de Gran Canaria, Las Palmas de Gran Canaria, Spain, 20 Medical Oncology Department, Hospital Universitario Marqués de Valdecilla, IDIVAL, Santander, Spain, 21 Medical Oncology Department, Hospital San Pedro, Logroño, Spain, 22 Medical Oncology Department, Hospital Universitario Clínico San Carlos Madrid, Spain, 23 Medical Oncology Department, Hospital Universitario Santa Creu i Sant Pau, Barcelona, Spain, 24 Medical Oncology Department, Hospital Universitario Nuestra Señora de la Candelaria, Tenerife, Spain, 25 Medical Oncology Department, Hospital Universitario Fundación Alcorcón, Madrid, Spain, 26 Medical Oncology Department, Catalan Institute of Oncology, L'Hospitalet, Spain, 27 Medical Oncology Department, Hospital Universitario Vall d'Hebron, Barcelona, Spain, 28 Medical Oncology Department, Complejo Asistencial Universitario de Salamanca, IBSAL, Salamanca, Spain, 29 Medical Oncology Department, Hospital Universitario Central de Asturias, ISPA, Oviedo, Spain, 30 Hematology and Medical Oncology Department, Hospital Universitario Morales Meseguer, University of Murcia, IMIB, Murcia, Spain

\* j.gallegoplazas@gmail.com



**Data Availability Statement:** All relevant data are within the paper and its Supporting Information files. Additional data are accessible in case of need in the Agamenon registry.

## Abstract

### Background

Second-line treatments boost overall survival in advanced gastric cancer (AGC). However, there is a paucity of information as to patterns of use and the results achieved in actual clinical practice.

**Funding:** The authors received no specific funding for this study.

**Competing interests:** Dr. Javier Gallego declares advisory role for Amgen, Bayer, BMS, Ipsen, Lilly, Merck, Roche, Servier, and travel grants from Novartis, Amgen. No other authors have competing interests to declare. This does not alter our adherence to PLOS ONE policies on sharing data and materials.

## Materials and methods

The study population comprised patients with AGC in the AGAMENON registry who had received second-line. The objective was to describe the pattern of second-line therapies administered, progression-free survival following second-line (PFS-2), and post-progression survival since first-line (PPS).

## Results

2311 cases with 2066 progression events since first-line (89.3%) were recorded; 245 (10.6%) patients died during first-line treatment and 1326/2066 (64.1%) received a second-line. Median PFS-2 and PPS were 3.1 (95% CI, 2.9–3.3) and 5.8 months (5.5–6.3), respectively. The most widely used strategies were monoCT (56.9%), polyCT (15.0%), ramucirumab+CT (12.6%), platinum-reintroduction (8.3%), trastuzumab+CT (6.1%), and ramucirumab (1.1%). PFS-2/PPS medians gradually increased in monoCT, 2.6/5.1 months; polyCT 3.4/6.3 months; ramucirumab+CT, 4.1/6.5 months; platinum-reintroduction, 4.2/6.7 months, and for the HER2+ subgroup in particular, trastuzumab+CT, 5.2/11.7 months. Correlation between PFS since first-line and OS was moderate in the series as a whole (Kendall's $\tau$ = 0.613), lower in those subjects who received second-line (Kendall's $\tau$ = 0.539), especially with ramucirumab+CT (Kendall's $\tau$ = 0.413).

## Conclusion

This analysis reveals the diversity in second-line treatment for AGC, highlighting the effectiveness of paclitaxel-ramucirumab and, for a selected subgroup of patients, platinum reintroduction; both strategies endorsed by recent clinical guidelines.

## Introduction

Advanced gastric cancer (AGC) is the third leading cause of cancer death worldwide [1]. Chemotherapy (CT) is capable of improving overall survival (OS) and quality of life for individuals with AGC compared to best supportive care (BSC) [2]. In first line, platin-fluoropyrimidine schedules are the most widely recommended option [3], whereas the standard of care is the combination of trastuzumab and cisplatin-fluoropyrimidine for tumors that amplify or over-express human epidermal growth factor receptor-2 (HER2+) [4]. The benefit of first-line is limited; up to 25%-30% display progression at their first evaluation of response [5] and median progression-free survival (PFS) is 4–7 months [2], with approximately 50% of patients in suitable conditions to receive second-line treatment after progression since first-line [6, 7].

Numerous drugs have proven activity in second-line for AGC [8, 9]. Thus, a small randomized trial (NCT00144378) confirmed for the first time that the use of irinotecan vs BSC in second line discreetly prolonged OS [6]. In the COUGAR-2 study, docetaxel incremented OS *versus* BSC and likewise demonstrated a benefit in quality of life [10]. Both drugs again improved OS compared to BSC in a phase III trial [7], while the WJOG-4007 study detected no differences between them or between paclitaxel and irinotecan [9]. More recently, the use of ramucirumab plus paclitaxel vs paclitaxel in second line was seen to increase OS in all subgroups in the RAINBOW trial [11]. For its part, the REGARD study corroborated a gain in OS with ramucirumab vs BSC [12]. Both studies with ramucirumab were bolstered by favorable

quality of life analyses, as well as real-world data [13–15]. This positions ramucirumab as the recommended second-line strategy, whether in combination or monotherapy [16]. There are minimal data concerning how the use of the various alternatives available for second-line treatment has evolved, in addition to their efficacy in actual clinical practice [17].

Moreover, pembrolizumab has demonstrated efficacy in a second line study of carcinoma of the esophagus and of the gastroesophageal junction, in the pre-specified subgroup of PDL1-CPS≥10 [18], while efficacy in second-line was unproven for advanced gastric or gastro- gastroesophageal junction adenocarcinoma in the KEYNOTE-061 phase III study [19]. Treatment in second and successive lines for HER2+ tumors does not currently differ from the rest, given the absence of evidence in favor of anti-HER2 therapy [20, 21]. Nevertheless, these tumors are molecularly dissimilar.

Based on retrospective analysis, certain individuals who do not receive first-line treatment until progression might profit from reintroducing platin-fluoropyrimidine doublets, when the treatment-free interval exceeds three months [22]. This subgroup of patients is excluded from most recent second-line clinical trials for AGC [11, 12], and most updated clinical guidelines consider reintroduction of the first-line to be an appropriate alternative [16].

Likewise, treatment options with proven efficacy exist in various third-line scenarios [23–25]. This availability of options beyond first line makes survival susceptible to the outcomes associated with successive lines of treatment, which could have implications when designing clinical trials. We must therefore revisit the value of intermediate endpoints, such as PFS, as surrogates for OS [26–30].

Against this backdrop, we have conducted this study to evaluate patterns of use and outcomes related to each type of strategy in second line and the surrogate function of PFS in an AGC registry (AGAMENON).

## Material and methods

### Patients and design

The patient population assessed derive from the Spanish AGAMENON registry that enlists the collaboration of 34 Spanish hospitals and one center in Chile and recruits consecutive cases of unresectable or metastatic, locally advanced adenocarcinoma of the stomach, gastroesophageal junction, or distal esophagus [31–39].

Eligibility criteria are: patients with AGC, aged >18 years, who received first-line treatment with polyCT routinely administered in clinical practice (two- or three-agent schedule, with or without platin) [40], and experienced tumor progression or died during first-line treatment. Those cases that had completed neoadjuvant or adjuvant treatment before 6 months were excluded.

Data are managed through a website (http://www.agamenonstudy.com/) consisting of filters and a query-generating system to guarantee reliability and control missing and inconsistent data, as well as errors. Telephone and on-line monitoring (PJF) further guarantee data quality.

### Objectives

The primary objective of this study was to describe the pattern of second-line therapies administered from 2008 onward and the associated outcomes. The secondary objective was to assess the correlation between PFS and OS over time, in terms of clinical-pathological variables and use of second lines.

## Variables

Post-progression survival (PPS) and PFS-2 were defined as the time between initiation of second-line and all-cause mortality or progression, respectively, censuring those event-free individuals at the time of the last follow up. OS and PFS-1 were defined as the interval between commencement of first-line treatment and death for any cause or progression, respectively, censoring at last follow up.

Second-line strategies were categorized as: (1) platinum-reintroduction, defined as providing a second-line platin-based schedule to individuals who had received platin in first-line with no evidence of progression when plantin was stopped; (2) maintenance of trastuzumab post-progression, consisting of changing the backbone of CT in progression to first-line without discontinuing trastuzumab; (3) regimens containing ramucirumab included use of paclitaxel, irinotecan, or other cytotoxics in combination with ramucirumab; (4) ramucirumab in monotherapy; additionally, patients could receive (5) monoCT or (6) polyCT.

## Statistics

Survival functions were Kaplan-Meier estimates. Correlation between PFS and OS was quantified by Kendall's $\tau$ associated with Clayton's copula models for bivariate survival data. Sensitivity to second line was evaluated using multivariable binary logistic regression (covariates were HER2 status, histological subtype, signet ring cells, hepatic tumor burden, number of metastatic sites, and interval of time since withdrawal of platin in first-line). Continuous variables were analyzed by means of restricted cubic splines. Treatment effect was appraised using a Cox multivariable proportional hazards model. No data-driven criteria were used for the model specification. The covariates for the multivariable model were chosen by theoretical considerations, as recommended in the literature [41]. Thus, ECOG PS ($<2$, $\geq 2$), Lauren's histopathological subtype (intestinal, diffuse), number of metastatic sites ($\leq 2$, $>2$), liver tumor burden ($\leq 50$, $>50\%$), HER2 status (negative, positive), PFS-1, and best response to first-line (complete or partial response, stable disease, progressive disease) were used as confounding factors. Restricted cubic splines were used to model the non-linear effect of PFS-1. Given that it is an observational, fixed sample size study, inferences should be interpreted in accordance with the magnitude of the CI with a descriptive purpose (hypothesis generator). Analyses were performed with the R v3.1.6 software package, with rms and Copula.surv libraries [42, 43].

## Ethics statement

This study has Compliance with Ethical Standards. This study was approved in November, 4 th 2014 by the Ethics, Research and Investigation Committee in Hospital Morales Meseguer, Murcia, Spain. The Research Ethics Committee from Morales Meseguer General University Hospital first, and then all the rest of Autonomous Communities and participating hospitals approved the study. The Spanish Agency of Medicines and Medical Devices categorized this study as a post-marketing, prospective follow-up study. In every alive prospective or retrospective registered patient, written informed consent was obtained in order to be included in the study. Participants who were not alive at data collection had previously provided written informed consent to use their medical records for the purposes of research. This was carried out according to the requirements stated in the international guidelines regarding carrying out epidemiological studies and put forth in the International Guidelines for Ethical Review of Epidemiological Studies (Council for the International Organizations of Medical Sciences–CIOMS-, Geneva, 1991), as well as the Declaration of Helsinki (Seoul revision, October, 2008). This document defines the principles that must be scrupulously respected by any and all personas involved in the research. The treatment, communication, and conveyance of the personal data of all participants was adapted to

the Organic Law 3/2018, dated December 5, regarding the Protection of Personal Data requiring approval by a Clinical Research Ethics Board (CREB).

# Results

## Patients and second-line treatments

At the time of analysis, 2311 cases had been recorded that met eligibility criteria, 2066 progression events since first-line (89.3%) and 2103 deaths (90.9%). Of the latter, 245 (10.6%) died during first-line. Median PFS-1 was 5.6 months (95% CI, 5.5–5.9), while median OS was 10.2 months (95% CI, 9.8–10.7).

Second-line therapy was given to 1326/2066 (64.1%); 366 (17.7%) received three lines, and 98 (4.7%), four or more. Baseline characteristics are summarized in Table 1. The most common strategies were: monoCT 56.9% (n = 755), polyCT 15.0% (n = 199), ramucirumab+CT 12.6% (n = 167), platinum-reintroduction 8.3% (n = 110), trastuzumab-continuing schedules 6.1% (n = 81), and ramucirumab monotherapy 1.1% (n = 14). S1 Table displays characteristics per strategy used.

In subjects treated with platinum-reintroduction, the reason for discontinuing platinum in first-line before progression was: having completed the number of cycles established by their center's protocol (71.8%), toxicity (18.2%), patient request (2.7%), and other reasons (7.3%).

Of the participants who received trastuzumab in second-line, 14/81 (17.3%) had not received it in first-line. Trastuzumab was withheld from those 14 patients in first-line because their HER2 status was unavailable (7 cases); due to cardiac comorbidity (n = 3), or oncologist's decision (n = 4). S2 Table shows the data of use of these strategies by HER2 status.

Fig 1 illustrates the usage trend of these strategies over time, revealing that the only one with an upward trend is the incorporation of ramucirumab from 2012 onward.

## Response rate to second lines

The response rate to second lines was 12.7% (n = 168); 28.5% had stable disease (n = 378) and the disease control rate (response or stable disease) was 41.2%. Progression occurred in 55.1% (n = 731) and information regarding response was unavailable for 3.7% (n = 49) of the cases. Fig 2 illustrates response rates by second-line strategy and HER2 status. For descriptive purposes, the probability of response to second-line has been represented depending on histopathological subtype, prior response to first-line, HER2 status and platin-free interval (S1–S3 Figs; S3 Table). The underlying model suggests differences according to these features. For instance, in diffuse tumors not responding previously to platin, the odds of achieving response to ramucirumab+CT vs monoCT increased as a function of platin-free interval: odds ratio (OR) 1.53 (95% CI, 0.69–3.72) at one month; OR 2.22 (95% CI, 1.30–3.81) at three months, and OR 2.90 (95% CI, 1.41–5.97) at six months. Plots with the probability of response for HER2+ and HER2-negative tumors can be seen in S2 and S3 Figs, respectively.

## Survival endpoints in second lines

At the time of analysis, 93.7% had suffered a progression event and 86.2% died after second-line. Median PFS-2 and PPS were 3.1 (95% CI, 2.9–3.3) and 5.8 months (95% CI, 5.5–6.3), respectively. Fig 3 presents survival for both endpoints.

Survival endpoints for each treatment group are laid out in Table 2. The highest median PFS-2 and PPS were obtained with platinum-reintroduction: 4.2 (95% CI, 3.3–5.0) and 6.7 months (95% CI, 5.5–10.2) and with ramucirumab+CT: 4.1 (95% CI, 3.4–5.2) and 6.5 months (95% CI, 5.5–8.7), respectively. In the case of HER2+ tumors, trastuzumab-containing regimens achieved a median PFS-2 of 4.8 months (95% CI, 3.6–5.7) and PPS of 10.5 months (95%

**Table 1. Characteristics at the time of diagnosis.**

| Variables | Total, n (%), n = 2311 | Patients receiving 2nd-line, n (%), n = 1326 |
|---|---|---|
| Age, median (range) | 64 (20–89) | 63 (20–86) |
| Sex, female | 672 (29.1) | 370 (27.9) |
| Lauren subtype | | |
| Diffuse | 745 (32.2) | 409 (30.8) |
| Intestinal | 991 (42.8) | 589 (44.4) |
| Mixed | 107 (4.6) | 61 (4.6) |
| Not Available | 468 (20.2) | 267 (20.1) |
| Signet ring cells | 657 (28.4) | 354 (26.7) |
| HER2-positive | 502 (21.7) | 318 (23.9) |
| ECOG-PS basal | | |
| 0 | 533 (23.2) | 361 (27.2) |
| 1 | 1457 (63.0) | 849 (64.0) |
| ≥2 | 321 (12.8) | 114 (8.8) |
| Tumor stage at diagnosis, locally advanced unresectable | 134 (18.1) | 74 (5.5) |
| Histological grade | | |
| *1* | 225 (9.7) | 152 (11.5) |
| *2* | 628 (27.2) | 361 (27.2) |
| *3* | 933 (40.4) | 525 (39.6) |
| *Not available* | 525 (22.7) | 288 (21.7) |
| First-line treatment | | |
| *Anthracycline-based* | 464 (20.1) | 285 (21.5) |
| *Cisplatin-based doublet* | 472 (20.4) | 302 (22.8) |
| *Docetaxel-based* | 276 (11.9) | 154 (11.6) |
| *Irinotecan-based* | 43 (1.9) | 25 (1.9) |
| *Oxaliplatin-based* | 911 (39.4) | 498 (37.6) |
| *Other* | 145 (6.3) | 62 (4.7) |
| Metastases sites | | |
| *Ascites* | 545 (23.6) | 289 (21.8) |
| *Peritoneal* | 1011 (43.7) | 559 (42.2) |
| *Bone* | 235 (10.2) | 112 (8.4) |
| *Lung* | 308 (13.3) | 185 (14.0) |
| *Liver* | 876 (37.9) | 522 (39.4) |
| Burden of liver disease >50% | 453 (19.6) | 247 (18.6) |
| Number of metastases >2 | 629 (27.2) | 332 (25.0) |
| Primary tumor site | | |
| *Esophagus* | 183 (7.9) | 113 (8.5) |
| *GEJ* | 306 (13.2) | 166 (12.5) |
| *Stomach* | 1822 (78.8) | 1046 (79.0) |
| PFS-1 | 5.6 (5.4–5.9) | 6.8 (6.5–7.1) |
| Best response to first-line | | |
| *Complete response* | 22 (1.0) | 18 (1.4) |
| *Partial response* | 661 (28.6) | 465 (35.1) |
| *Stable disease* | 1028 (44.5) | 570 (43.0) |
| *Progression disease* | 600 (26.0) | 273 (20.6) |

Abbreviations: ECOG-PS, Eastern Cooperative Group Performance Status; GEJ, gastroesophageal junction.

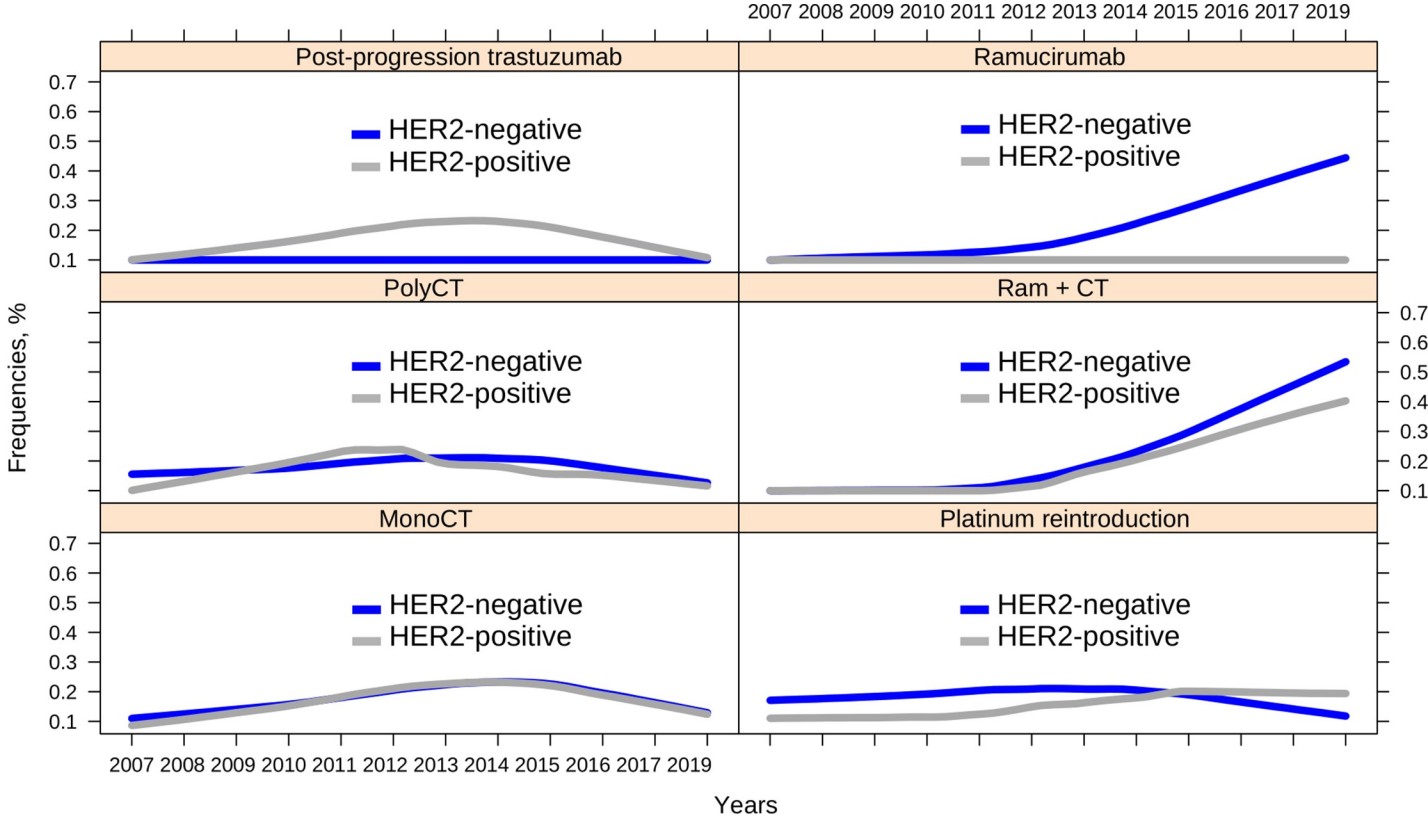

**Fig 1. Time trends in the use of second-line schedules based on HER2 status.** Abbreviations: polyCT, polychemotherapy; monoCT, monochemotherapy; Ram+CT, ramucirumab+chemotherapy.

CI, 5.5–12.1). In a sensitivity analysis, after excluding 14 subjects without first-line trastuzumab, the remaining patients obtained a similar median PFS-2/PPS, 4.80 (CI 95%, 3.45–5.75) and 10.8 months (CI 95%, 7.1–14.6) respectively. MonoCT yielded the worst results with median PFS-2 of 2.6 months (95% CI, 2.4–2.7) and PPS of 5.1 months (95% CI, 4.6–5.7). In the multivariable Cox model, taking monoCT as reference, ramucirumab+CT (HR 0.62; 95% CI, 0.51–0.74), platinum-reintroduction (HR 0.76; 95% CI, 0.61–0.94), polyCT (HR 0.81; 95% CI, 0.69–0.96), and trastuzumab+CT (HR 0.58, 95% CI; 0.44–0.77, in HER2+) were associated with better PFS-2. The data as per HER2 subtype are detailed in Table 2 and Fig 4.

### Correlation of PFS & OS with each treatment strategy

The correlation between PFS-1 and OS is moderate in the complete series (n = 2311, Kendall's τ = 0.613), lower in individuals who received a second-line (Kendall's τ = 0.539). The possibility of having effective second lines available dilutes the surrogate value of PFS-1, principally in individuals who receive CT-ramucirumab. Correlations for each treatment strategy are as follows: ramucirumab+CT (Kendall's τ = 0.413), polyCT (Kendall's τ = 0.503), monoCT (Kendall's τ = 0.539), trastuzumab+CT (Kendall's τ = 0.566), and platinum-reintroduction (Kendall's τ = 0.585) (S4 Table).

### Discussion

Within the context of AGC, second-line therapy has been proven to enhance OS compared to BSC to a statistically significant extent [44]. In a meta-analysis of 10 clinical trials, polyCT was

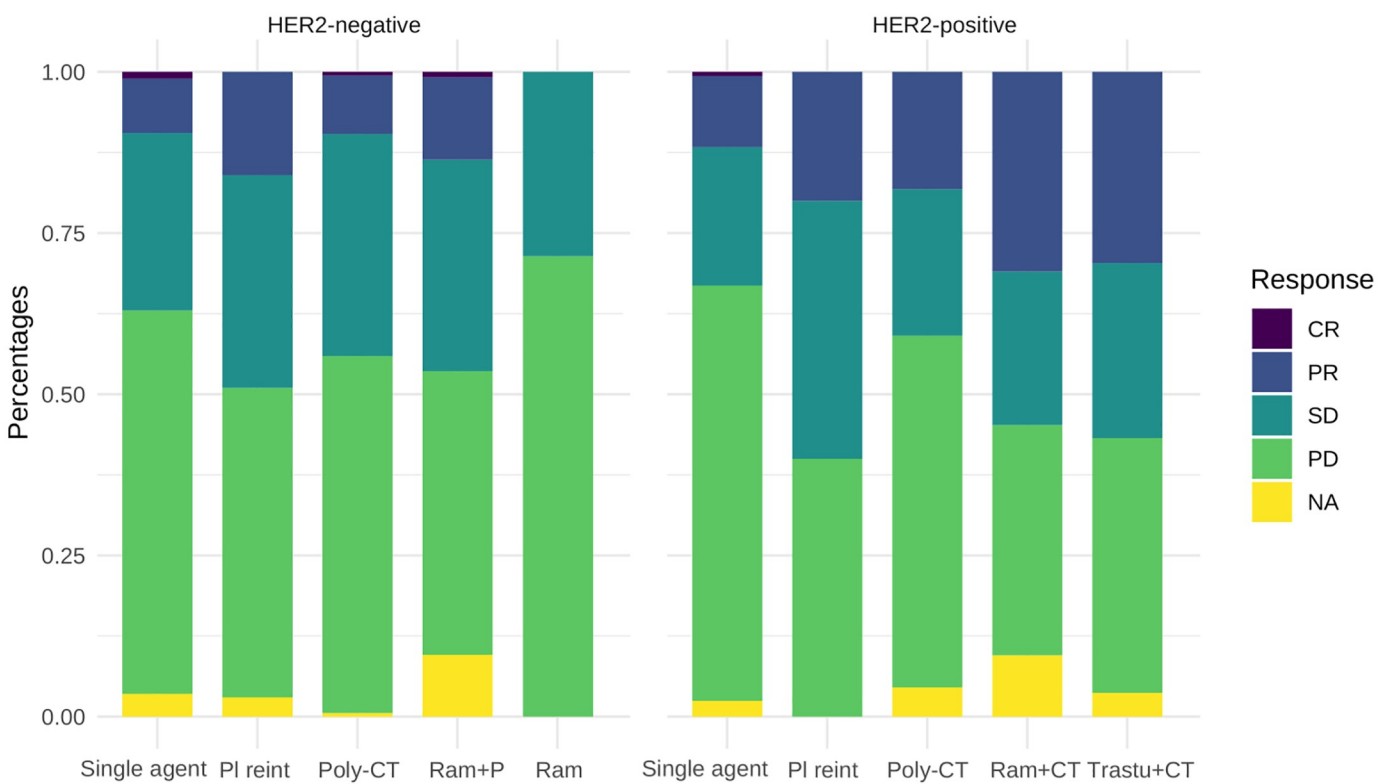

**Fig 2. Response rates according to HER2 and strategy.** Abbreviations: Pl reint, platinum reintroduction; poly-CT, polychemotherapy; P, paclitaxel; Ram, ramucirumab; CT, chemotherapy; Trastu, trastuzumab; CR, complete response; PR, partial response; SD, stable disease; PD, progressive disease; NA, not available. *Paclitaxel was the cytotoxic used in all patients with HER-negative tumors who received ramucirumab+chemotherapy, whereas paclitaxel and other cytotoxics were associated with ramucirumab in HER+ tumors.

more effective than monoCT [45], while a network meta-analysis suggests that the combination of paclitaxel+ramucirumab is most likely to be the best schedule available to date [46]. However, data with reference to real-world use of second lines (without the usual clinical trial selection biases) are scant. Moreover, there is a paucity of information about the strategies clinicians apply pragmatically, such as platinum-reintroduction or using trastuzumab beyond progression.

To investigate these aspects, we evaluated the use of second-line in the 64.1% of the AGC registry patients who received it, a percentage similar to that observed by other authors [6, 7]. The individuals who received second-line tended to be those who had benefitted most from first-line, with longer PFS-1.

Our data corroborate that polyCT and CT+ramucirumab is superior to monoCT in daily practice [45, 46]. Bearing in mind the safety profile of each strategy in indirect comparisons, and the available scientific evidence, this would endorse the established role of ramucirumab+-paclitaxel as the current standard of second-line treatment in AGC. The AGAMENON data endorse this consideration, by revealing a trend toward increased use of ramucirumab, alone and in combination, compared to the remaining second-line strategies, which are declining.

Furthermore, the study indicates that histopathological subtype, therapy administered, time since platin withdrawal, and better response to first-line might be among the factors associated with response. In particular, chemosensitivity to second-line are continuously and non-

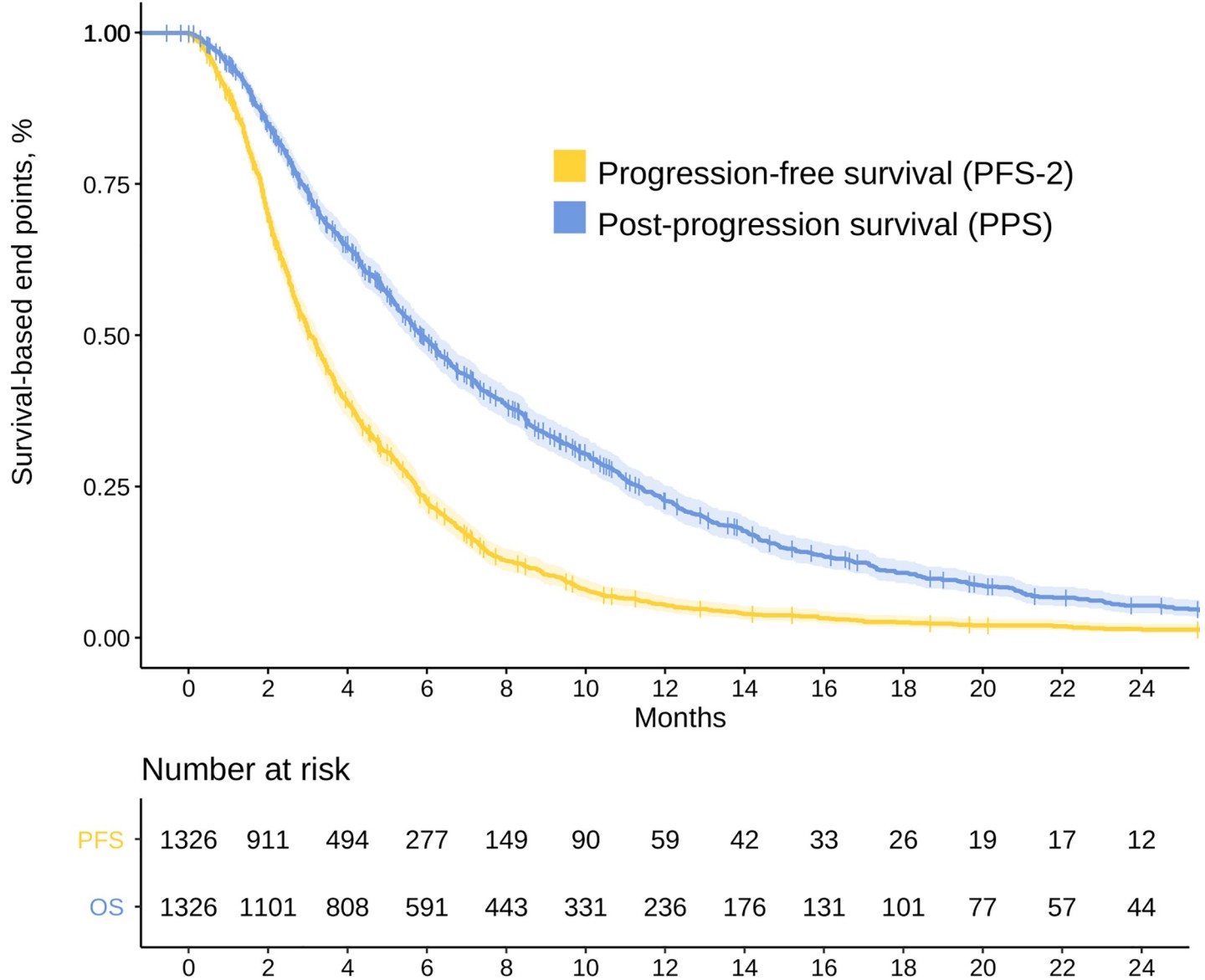

**Fig 3. Survival curves for PFS-2 and PPS (n = 1326).** Abbreviations: PFS-2, progression-free survival to second-line of treatment; PPS, post-progression survival.

linearly related to the platin-free period. PFS-1 is a known and consistent predictive factor during second-line therapy [44, 47]. Diffuse tumors are more refractory to second-line treatment than the intestinal subtype, although this depends on the interaction with the platin-free interval. Thus, even in adverse scenarios, such as treatment-resistant diffuse tumors, the probability of response is twofold in those exposed to ramucirumab+CT vs monoCT, indicating that treatment choice is key to achieving benefit.

Based on retrospective analysis, reintroduction of the same drug combination should be contemplated for patients in whom first-line treatment was discontinued and time to progression exceeded three months, provided that any toxicity issues have been resolved [22] and as recommended in the most recently updated guidelines [16]. In this registry, the reintroduction of firs-line platin-based therapy (10% of Her2- patients) was associated with the highest disease control rate and median PPS. These results are comparable to those of the study by Okines

**Table 2. Survival endpoints based on the strategy for HER+ and HER-negative tumors.**

| Variables | n/events | Median PFS-2, months (95% CI) | n/events | Median PPS, months (95% CI) |
|---|---|---|---|---|
| **All** | | | | |
| Mono-CT | 755/723 | 2.6 (2.4–2.7) | 755/677 | 5.1 (4.6–5.7) |
| Poly-CT | 199/194 | 3.4 (2.7–3.9) | 199/172 | 6.3 (5.6–7.2) |
| Ram-CT | 167/139 | 4.1 (3.4–5.2) | 167/104 | 6.5 (5.5–8.7) |
| Plat reintroduction | 110/104 | 4.2 (3.3–5.0) | 110/99 | 6.7 (5.5–10.2) |
| **HER2-negative** | | | | |
| Ram | 14/11 | 2.8 (1.8-NA) | 14/10 | 5.0 (3.0-NA) |
| Mono-CT | 592/566 | 2.6 (2.4–2.8) | 592/527 | 4.9 (4.3–5.4) |
| Poly-CT | 177/172 | 3.4 (2.7–4.8) | 177/157 | 6.2 (5.5–7.1) |
| Ram-CT | 125/104 | 3.8 (3.3–5.1) | 125/84 | 6.5 (5.1–9.4) |
| Plat reintroduction | 100/95 | 4.1 (3.2–4.8) | 100/91 | 6.6 (5.4–9.8) |
| **HER2-positive** | | | | |
| Mono-CT | 163/157 | 2.7 (2.4–3.2) | 163/150 | 6.7 (5.2-NA) |
| Poly-CT | 22/22 | 3.0 (2.5–5.7) | 22/20 | 8.6 (5.0–14.9) |
| Ram-CT | 42/35 | 4.7 (3.2–6.3) | 42/30 | 7.3 (5.5–12.1) |
| CT + Trastuzumab | 81/72 | 4.8 (3.6–5.7) | 81/66 | 10.5 (5.5–12.1) |
| Plat reintroduction | 10/9 | 5.2 (3.1-NA) | 10/8 | 11.7 (7.3–13.3) |

Abbreviations: Ram, ramucirumab; Plat, platinum; CT, chemotherapy; PFS-2, progression-free survival to second-line of treatment; PPS, post-progression survival.

*et al* that revealed that the reintroduction of platin was associated with median PFS-2 and PPS of 3.9 and 6.6 months, respectively [22], depending on prior chemosensitivity to platin, platin-free interval, and histological subtype. Therefore, given that platin is sometimes discontinued due to cumulative toxicity, proceeding with fluoropyrimidine until progression [48], platin reintroduction might be an especially useful option in intestinal tumors, sensitive to platin in first-line, with a prolonged platin-free interval and in the absence of residual toxicity.

Another strategy arising in this real world evidence analysis is that of using trastuzumab following progression, which in this registry accounts for 25.5% of HER+ tumors, although evidence for trastuzumab in second-line treatment of AGC is lacking [49, 50]. Likewise, our data corroborate the favorable prognostic effect of HER2+ status that is maintained beyond first-line [34]. Still, the reader must be mindful of the current lack of positive results in clinical trials that have assessed anti-HER2 therapy in second-line [20, 21], as well as the confirmed benefit of ramucirumab in cases in which trastuzumab was administered in first-line [51].

Finally, we have examined the surrogate function of PFS-1 within the context of the availability of treatment strategies after first-line. OS has traditionally been the gold-standard endpoint in clinical trials of first-line therapy for AGC; nonetheless, PFS continues to be routine in various randomized AGC studies [26–28]. The advantages of PFS include shortened study duration, smaller sample size, and the absence of interference of post-progression therapies. Overall, the use of intermediate endpoints calls for statistical proof of the validity of the surrogate, as well as the justification of the clinical value that delaying progression has for the patient's quality of life [52]. Our data reveal that the correlation between PFS-1 and OS is moderate in actual practice, with a magnitude slightly lower than that reported in the literature [53]. In fact, the possibility of having effective second lines available dilutes the surrogate value of PFS-1, principally in individuals who receive CT-ramucirumab. Bearing in mind the gradual increase in the use of ramucirumab in our series, this would call into question the appropriateness of substituting OS for PFS-1 in studies of first line in AGC.

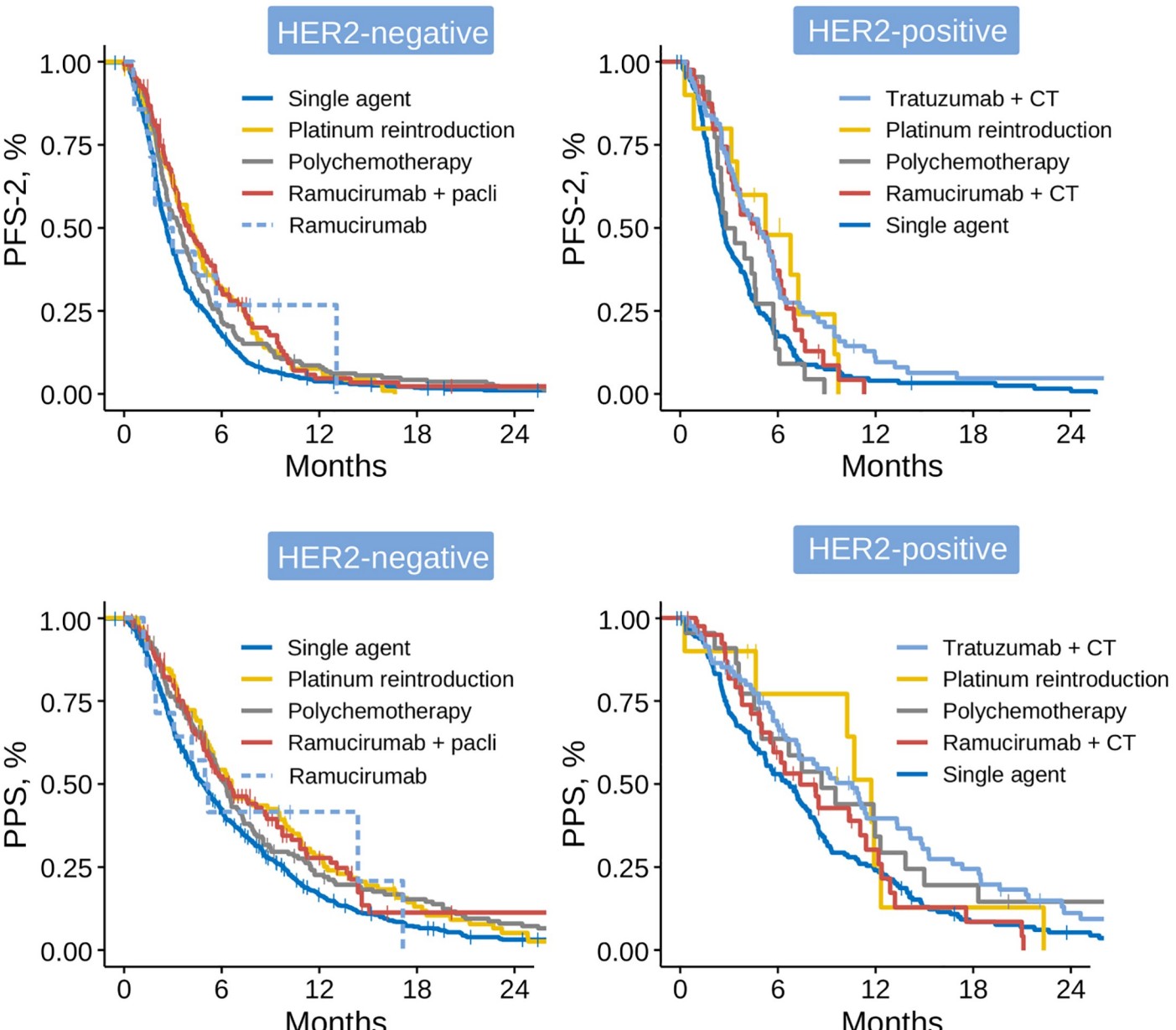

**Fig 4. Survival functions since initiation of second-line by HER2 status and treatment strategy.** Abbreviations: PFS-2, progression-free survival to second-line of treatment; PPS, post-progression survival; CT, chemotherapy; Pacli, paclitaxel.

There are several limitations implicit in observational studies, such as this one, in which the criteria that mediate in the decisions regarding second lines depend on the evolution of the disease that are not initially present and, as such, are difficult to capture in a registry of this kind. In addition, patients in this registry received first-line treatment with polyCT, excluding more fragile patients who were only candidates for monoCT. Nevertheless, survival endpoints and baseline characteristics are adequately typified through regular reviews and updating of the information.

## Conclusion

In short, our study provides the largest real world practice data set regarding the use of second lines for AGC, backing up the scientific evidence derived from previous clinical trials and smaller retrospective analyses. Our analysis reveals the diversity in second-line treatment for AGC, highlighting the effectiveness of paclitaxel-ramucirumab and, for a selected subgroup of patients, platinum reintroduction; both strategies endorsed by recent clinical guidelines. Additionally, it disputes the role of PFS as a surrogate for OS with the progressive incorporation of more efficacious strategies in successive lines of treatment.

## Supporting information

**S1 Table. Characteristics at the time of diagnosis by second-line strategy.**
(DOCX)

**S2 Table. Chemotherapy regimens used based on HER2 status.**
(DOCX)

**S3 Table. Response rate depending on HER2 status and treatment strategy.**
(DOCX)

**S4 Table. Correlation between progression-free survival to first-line of treatment and overall survival.**
(DOCX)

**S1 Fig. Probability of response to second-line depending on platinum-free interval, response to first-line, and subtype (all patients).**
(TIFF)

**S2 Fig. Probability of response to second-line depending on platinum-free interval, response to first-line, subtype and second-line strategy (HER2-negative).**
(TIFF)

**S3 Fig. Probability of response to second-line depending on platinum-free interval, response to first-line, and second-line strategy (HER2-positive subset).**
(TIFF)

## Acknowledgments

We thank Priscilla Chase Duran for editing the manuscript, Natalia Cateriano, Miguel Vaquero, and IRICOM S.A. for supporting the registry website. We are indebted to all patients, as well as to AGAMENON centres and investigators who particpated in this research and made it possible.

## Disclaimers

**iii.** AGAMENON registry is part of the Evaluation of Results and Clinical Practice Section included in the Spanish Society of Medical Oncology (SEOM).

## Author Contributions

**Conceptualization:** Almudena Cotes Sanchís, Javier Gallego, Paula Jimenez-Fonseca, Alberto Carmona-Bayonas.

**Data curation:** Almudena Cotes Sanchís, Javier Gallego, Raquel Hernandez, Virginia Arra-zubi, Ana Custodio, Juana María Cano, Gema Aguado, Ismael Macias, Carlos Lopez, Flora López, Laura Visa, Marcelo Garrido, Nieves  Martínez Lago, Ana Fernández Montes, María Luisa Limón, Aitor Azkárate, Paola Pimentel, Pablo Reguera, Avinash Ramchandani, Juan Diego Cacho, Alfonso Martín Carnicero, Mónica Granja, Marta Martín Richard, Carolina Hernández Pérez, Alicia Hurtado, Olbia Serra, Elvira Buxo, Rosario Vidal Tocino, Paula Jimenez-Fonseca, Alberto Carmona-Bayonas.

**Formal analysis:** Alberto Carmona-Bayonas.

**Investigation:** Javier Gallego, Paula Jimenez-Fonseca, Alberto Carmona-Bayonas.

**Methodology:** Almudena Cotes Sanchís, Javier Gallego, Paula Jimenez-Fonseca, Alberto Car-mona-Bayonas.

**Project administration:** Paula Jimenez-Fonseca.

**Supervision:** Almudena Cotes Sanchís, Javier Gallego, Paula Jimenez-Fonseca, Alberto Car-mona-Bayonas.

**Validation:** Javier Gallego, Paula Jimenez-Fonseca, Alberto Carmona-Bayonas.

**Writing – original draft:** Javier Gallego, Paula Jimenez-Fonseca, Alberto Carmona-Bayonas.

**Writing – review & editing:** Almudena Cotes Sanchís, Javier Gallego, Paula Jimenez-Fonseca, Alberto Carmona-Bayonas.

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
