## [Decision Letter · Decision Letter 0]

29 May 2020

PONE-D-20-09864

Second-line treatment in advanced gastric cancer: data from the Spanish AGAMENON registry

PLOS ONE

Dear Dr. Gallego Plazas,

Thank you for submitting your manuscript to PLOS ONE. After careful consideration, we feel that it has merit but does not fully meet PLOS ONE’s publication criteria as it currently stands. Therefore, we invite you to submit a revised version of the manuscript that addresses the points raised during the review process.

We look forward to receiving your revised manuscript.

Kind regards,

Academic Editor

PLOS ONE

Additional Editor Comments:

In general the paper was well-written, a lot of efforts were spent on it. In addition to addressing the reviewers' concerns, you have to emphasize the main outstanding points of your paper that will catch the readers attention to your paper. Emphasize the points what this retrospective analysis adds or confirms to current knowledge. Consequently you should re-write the conclusion part of the paper since there is no take home points or important findings there. The last para of discussion in unnecessary please delete it (beginning with the readers must..). According to the heterogeneity of the treatment groups cox-regression analysis does not seem to be appropriate unless the including variables are appropriately described for this analysis, since there were lots variables (age, gender, tumor type, performance status, etc). Which model did you use for cox-regression?

Journal Requirements:

2. Please include your tables as part of your main manuscript and remove the individual files. Please note that supplementary tables should remain as separate "supporting information" files.

3. Please provide additional details regarding participant consent.

In the ethics statement in the Methods and online submission information, please ensure that you have specified whether participants who were not alive at data collection had previously provided written informed consent to use their medical records for the purposes of research.

4. Thank you for including your ethics statement:

'A multicenter Research Ethics Committee from all the Autonomous Communities and participating hospitals approved the study.'

(a) Please amend your current ethics statement to include the full name of the ethics committee/institutional review board(s) that specifically approved your specific study. 

(b) Once you have amended this/these statement(s) in the Methods section of the manuscript, please add the same text to the “Ethics Statement” field of the submission form (via “Edit Submission”).

5. Thank you for providing the following Funding Statement: 

"JG Phd. Declares advisory role for Amgen, Bayer, BMS, Ipsen, Lilly, Merck, Roche, Servier. Travel grants from Novartis, Amgen.

None of the funders played any role in the study design, data collection and analysis, decision to publish, or preparation of the manuscript."

We note that one or more of the authors is affiliated with the funding organization, indicating the funder may have had some role in the design, data collection, analysis or preparation of your manuscript for publication; in other words, the funder played an indirect role through the participation of the co-authors.

a. If the funding organization did not play a role in the study design, data collection and analysis, decision to publish, or preparation of the manuscript and only provided financial support in the form of authors' salaries and/or research materials, please review your statements relating to the author contributions, and ensure you have specifically and accurately indicated the role(s) that these authors had in your study in the Author Contributions section of the online submission form. Please make any necessary amendments directly within this section of the online submission form.  Please also update your Funding Statement to include the following statement: “The funder provided support in the form of salaries for authors [insert relevant initials], but did not have any additional role in the study design, data collection and analysis, decision to publish, or preparation of the manuscript. The specific roles of these authors are articulated in the ‘author contributions’ section.”

If the funding organization did have an additional role, please state and explain that role within your Funding Statement.

6. Please amend your manuscript to include your abstract after the title page.

7. Please upload a copy of Figure 4, to which you refer in your text on page 17. If the figure is no longer to be included as part of the submission please remove all reference to it within the text.

8. Please include captions for your Supporting Information files at the end of your manuscript, and update any in-text citations to match accordingly. Please see our Supporting Information guidelines for more information: http://journals.plos.org/plosone/s/supporting-information

9. Your ethics statement must appear in the Methods section of your manuscript. If your ethics statement is written in any section besides the Methods, please move it to the Methods section and delete it from any other section. Please also ensure that your ethics statement is included in your manuscript, as the ethics section of your online submission will not be published alongside your manuscript.

Reviewers' comments:

Reviewer's Responses to Questions

**Comments to the Author**

1. Is the manuscript technically sound, and do the data support the conclusions?

Reviewer #1: Yes

Reviewer #2: Partly

2. Has the statistical analysis been performed appropriately and rigorously? 

Reviewer #1: Yes

Reviewer #2: No

3. Have the authors made all data underlying the findings in their manuscript fully available?

Reviewer #1: Yes

Reviewer #2: Yes

4. Is the manuscript presented in an intelligible fashion and written in standard English?

Reviewer #1: Yes

Reviewer #2: Yes

5. Review Comments to the Author

Reviewer #1: Author describe and comparison the CTs for 2nd line GC patients from real-world data of Spain.

1. In multivariate analysis for survival, did you include the potential prognostic factor, for example PS, ALP, No of mets and others? As you mentioned, the patients who received mono CT as second line have different condition compared to polyCT.

2. Do you have a data of the patients who received ICI as 3rd and later line? Because the response of ICI may continue for a long time, the use of ICI may influence the survival of each groups.

3. In Asian countries, first line platinum agent continue more than 6 courses if the patients have no toxicities and inconvenient. The re-introduction of platinum depends of the strategy of using of first line platinum administration. How the first line platinum are used for the patients with GC in Spain?

Reviewer #2: Sanchis et al presented a retrospective/observational study on second line treatment options for advanced gastric cancer AGC according to clinical practice in several cancer centers in Spain and Chile. AGC, after first line therapy failure, has poor prognosis but recent developments has added to the clinical practice scenario, new therapeutic approaches with a clear benefit on survival. The addition of VEGFR2 antagonist ramucirumab to taxanes, for instance, increased OS compared to mono-chemotherapy regimens.

The topic addressed by this manuscript is important and the paper is well written, however it lacks novelty and simply provide a description of several regimens used in second-line setting throughout the last 12 years. It appears clear that the use of ramicurumab is climbing in the most recent time due to the wide spreading of this agents from 2015 when RAINBOW trial was published. Moreover, the methods appear to be sound but the correlation of PFS and OS with each treatment strategy in the results section is not easily readable and should be expanded and better performed to ensure that readers understand exactly what the researchers wanted to state. Thirdly, platinum re-introduction, when feasible, could be a valid therapeutic option, however the conclusion of the manuscript appears to be way stronger due to low number (8.3%), the paucity of patients that will be able to tolerate the reintroduction of platinum-doublet in this setting and the results, totally comparable to Ram-taxane, which shows a better toxicity profile.

Lastly, the dissertation on PFS as adequate surrogate of OS is interesting. This is a current unmet need in oncology research. Specifically, in this study, authors focused on PPS (post progression survival). According to their data, it that could have some relevance in Her2 positive AGC since the anti-her2 action of trastuzumab, even though a radiologic progression, keeps the ability to control the Her-2 enriched population which could lead to a longer overall survival. However, for Her2-negative tumors, the magnitude of the benefit is lower, reaching less than 3 months in the best subgroup. In this case, QoL has much more impact and relevance for patients and should be primarily assessed in studies on poor prognosis cancer after first setting

6. PLOS authors have the option to publish the peer review history of their article (what does this mean?). If published, this will include your full peer review and any attached files.

Reviewer #1: Yes: Ken Kato

Reviewer #2: No

---

## [Author Response · Author response to Decision Letter 0]

20 Jun 2020

Dear editor, 

In accordance with the Editor’s suggestions and concerns regarding the manuscript entitled, “Second-line treatment in advanced gastric cancer: data from the Spanish AGAMENON registry”, please find the new version enclosed.

On behalf of all the co-authors, I would like to thank the editor and the reviewers for their thoughtful and insightful comments on our work, which we have considered very closely, while preparing this revised version of the manuscript as detailed in the point-by-point reply named Response to Reviewers.

We trust that we have addressed all the issues raised by the reviewers to their satisfaction and you now find the manuscript suitable for publication in your journal. 

Thank-you very much for your consideration.

Respectfully,

MD PhD, Javier Gallego Plazas

---

## [Editor Report · Decision Letter 1]

24 Jun 2020

Second-line treatment in advanced gastric cancer: data from the Spanish AGAMENON registry

PONE-D-20-09864R1

Dear Dr. Gallego Plazas,

We’re pleased to inform you that your manuscript has been judged scientifically suitable for publication and will be formally accepted for publication once it meets all outstanding technical requirements.

Kind regards,

Academic Editor

PLOS ONE
---

## [Editor Report · Acceptance letter]

21 Jul 2020

PONE-D-20-09864R1 

Second-line treatment in advanced gastric cancer: data from the Spanish AGAMENON registry 

Dear Dr. Gallego:

I'm pleased to inform you that your manuscript has been deemed suitable for publication in PLOS ONE. Congratulations! Your manuscript is now with our production department. 

Kind regards, 

on behalf of

Dr. Hakan Buyukhatipoglu 

Academic Editor

PLOS ONE